# Low Sleep Hygiene Is Associated with Less Adherence to the Mediterranean Diet in Chilean Schoolchildren from Rural Public Schools—A Cross-Sectional Study

**DOI:** 10.3390/children10091499

**Published:** 2023-09-01

**Authors:** Rafael Zapata-Lamana, Jessica Ibarra-Mora, Fernanda Carrasco-Marín, Samuel Durán-Agüero, Jorge Cuevas-Aburto, Maria Antonia Parra-Rizo, Igor Cigarroa

**Affiliations:** 1Escuela de Educación, Universidad de Concepción, Los Ángeles 4440000, Chile; rafaelzapata@udec.cl; 2Departamento de Educación Física, Deportes y Recreación, Universidad Metropolitana de Ciencias de la Educación, Santiago 8330106, Chile; jessica.ibarra@umce.cl; 3Centro de Vida Saludable, Universidad de Concepción, Concepción 4030000, Chile; fercarrasco@udec.cl; 4Escuela de Nutrición y Dietética, Facultad de Ciencias para el Cuidado de la Salud, Universidad San Sebastián, Santiago 8330106, Chile; samuel.duran@uss.cl; 5Centro de Aprendizaje, Universidad Santo Tomás, Los Ángeles 4440000, Chile; jorgecuevas@santotomas.cl; 6Faculty of Health Sciences, Valencian International University (VIU), 46002 Valencia, Spain; 7Department of Health Psychology, Faculty of Social and Health Sciences, Campus of Elche, Miguel Hernandez University (UMH), 03202 Elche, Spain; 8Escuela de Kinesiología, Facultad de Salud, Universidad Santo Tomás, Los Ángeles 4440000, Chile

**Keywords:** sleep, diet, Mediterranean, adolescent, students, Chile

## Abstract

The Mediterranean diet stands as a widely acknowledged and health-promoting dietary pattern, renowned for its notable linkage to the mitigation of noncommunicable chronic maladies. Nonetheless, the existing body of evidence concerning the potential interrelation between sleep hygiene and this dietary regimen remains circumscribed. The main objective was to determine the association between sleep hygiene and adherence to the Mediterranean diet in Chilean schoolchildren from rural public schools in southern Chile. A non-experimental study was carried out, with an analytical, cross-sectional design. A total of 265 students (56.6% women, mean age 13.5 ± 1.8) from a rural community in southern Chile were recruited. Sleep habits were evaluated using Section 6 of the Life Habits and Adolescence Questionnaire, Sleep and Rest, and adherence to the Mediterranean diet was assessed with the KIDMED Mediterranean Diet Adherence Questionnaire. The main results indicated that 52.8% of schoolchildren need to improve adherence to the Mediterranean diet and 16.6% have a low-quality Mediterranean diet. A high percentage of schoolchildren have behaviors related to poor sleep hygiene (going to bed late (46%), waking up tired and wanting to continue sleeping (63.8%), and having problems falling asleep (42.6%)). Schoolchildren who got up after 8:30 a.m., those who fell asleep after midnight, upon conducting a comparative analysis of the students based on their sleep patterns, those who woke up tired and those who had trouble falling asleep had a lower level of adherence to the Mediterranean diet compared to schoolchildren who got up earlier than 8:30 a.m., fell asleep before midnight, did not wake up tired, and those who did not find it difficult to fall asleep, respectively. In conclusion, having poor sleep patterns including difficulties in both awakening and falling asleep are associated with less adherence to the Mediterranean diet in schoolchildren from rural public schools in southern Chile. Monitoring these variables and promoting healthy lifestyle habits within the educational community are essential measures.

## 1. Introduction

Sleep hygiene refers to the behaviors and environmental recommendations that contribute to adequate sleep, including appropriate sleep schedules, healthy sleep habits, a sleep-supportive environment, and various physiological practices that facilitate having good sleep [1]. Sleep hygiene, originally formulated for addressing mild to moderate insomnia, seeks to regulate the factors influencing the sleep patterns of children and adolescents [2,3]. There is substantial evidence that insufficient sleep and poor sleep quality in schoolchildren are associated with numerous problems, such as reductions in cognitive and academic performance, behavioral difficulties, challenges with daily functioning [4], and increased risk of overweight and obesity [5].

Recent studies have shown changes in sleep patterns among children and adolescents. In Chile, the National Health Survey revealed that 64.8% of young people reported having sleep problems. Specifically, 11.8% of secondary school students reported poor sleep quality. According to international evidence, our country is among the eight countries where schoolchildren and adolescents suffer the most sleep problems (63% of 4th-grade students and 74% of 8th-grade students) [6]. The evidence suggests that some of the causes of these problems are the decrease in parental control over their children’s sleep hours and, as evident in the last decade, the increase in technology and its growing accessibility, which have also had a negative impact [7].

In addition to changes in sleep patterns in recent years, excess weight in children and adolescents is a significant problem worldwide [8], and the situation is no different in Chile. According to the 2020 nutritional map, 54% of Chilean schoolchildren are overweight [9]. Although multiple factors have contributed to this epidemic, one of the main causes is lifestyle, specifically dietary patterns [10], screen exposure [11], and physical activity time [12].

Although several studies have linked sleep to body composition in children and adolescents, these associations are primarily based on the relationship between short and long sleep durations and BMI and adiposity [13]. However, the causality of this relationship is not entirely clear. Evidence has shown that short sleep duration leads to hormonal and metabolic alterations that trigger changes in food choices, energy intake, and macronutrient intake [14]. Conversely, it has been reported that specific food consumption such as vitamin D [15] and nutritional status impact sleep quality and quantity [16], suggesting that the association between these variables may be bidirectional. In this context, evidence suggests that an inflammatory diet is associated with lower sleep quality [17], while others consider that sociodemographic characteristics and dietary patterns may influence this relationship [18]. Accumulating scientific evidence suggests that dietary components possess the potential to influence and even predict sleep outcomes, both within individuals who are in good health and those presenting clinical conditions [19,20,21]. In this context, the Mediterranean diet, characterized by its plant-based composition rich in antioxidants and unsaturated fats, has consistently demonstrated an association with decreased prevalence of noncommunicable diseases and overall mortality, firmly establishing its standing as one of the healthiest dietary regimens [22]. There is no strict consensus on what constitutes a Mediterranean diet in percentages and macro-nutrients [23]. However, it is accepted that the Mediterranean dietary pattern includes moderate consumption of unsaturated fats, fish, lean meats, fruit, vegetables, nuts, legumes, and low consumption of red meat and saturated fats [24]. This dietary pattern is associated with multiple beneficial health outcomes, such as preventing cardiovascular disease, reduced risk of certain types of cancer, and even better cognitive and mental cognitive and mental health [25].

In general, it has been shown that foods rich in melatonin—a hormone fostering sleep—or its precursors, namely tryptophan and serotonin, along with essential micronutrients like vitamin D, B, magnesium, and zinc, as well as carbohydrate-rich fare and specific food items such as cherries and fish, exhibit the potential to enhance various sleep parameters (e.g., sleep latency, duration, efficiency) [26]. However, nutrients and foods are not consumed in isolation but in combination with dietary patterns. In this trajectory, the impact of dietary regimens, including the Mediterranean diet, on sleep hygiene remains a relatively less explored terrain [27,28]. However, in Latin American countries like Chile, it is still considered an emerging area, and there is limited evidence on how sleep hygiene may be related to nutrition. Its implications have been poorly explored in the school-aged population, particularly in rural contexts, which is relevant considering the high prevalence of sleep problems and obesity reported in Chilean children and how these variables impact school performance [29]. According to the consulted literature, this would be the first study to analyze the association between sleep hygiene and adherence to the Mediterranean diet in schoolchildren from rural public schools in southern Chile. Thus, the study objectives were to: (a) characterize sleep hygiene indicators, (b) analyze adherence to the Mediterranean diet, and (c) determine the association between sleep hygiene indicators and adherence to the Mediterranean diet in Chilean schoolchildren from rural public schools in southern Chile.

## 2. Materials and Methods

*Study Design*: The present study used a non-experimental, analytical, and cross-sectional design.

*Participants:* Students between 7th and 12th grade (aged 11 to 18 years) from a small rural community in southern Chile were invited to participate. The total enrollment in April 2021 was 1067 students. The recruitment took place during the COVID-19 pandemic, amid mobility restrictions and without in-person lessons. A total of 513 parental consents were obtained, of which 462 parents agreed to have their children participate. One hundred and five students did not complete all the questionnaires, and 92 students omitted some questions. Despite the communication and connectivity challenges, final data from 265 students (115 males and 150 females) were collected. Thus, a non-probabilistic sample of voluntary subjects was obtained. All students signed an assent and their parents consented and completed all three questionnaires.

*Procedure:* The project was developed based on the need to diagnose the current dietary pattern and sleep habits of high school students in a rural commune in southern Chile. This need arose after a 2020 school year in which in-person lessons were not held due to the COVID-19 pandemic. A search process for instruments that would provide the required information and align with the objectives was conducted. The documentation was then submitted to the Scientific Ethics Committee of the Central-South macrozone of Santo Tomás University (CEC-CS UST) and received approval in April of 2021, with code number 18-21.

Subsequently, the documentation was presented to the school management teams of the province, as well as the teachers responsible for the participating classes. They were informed about the online application process, the protocol for obtaining parental consent and student assent, and the procedures to be followed. During this phase, the school principals in charge of each school were trained on how to administer the questionnaires, reviewed the questions, provided guidance on how to assist students with any questions they might have, and coordinated actions for implementation.

In parallel, the instruments were transferred to a Google questionnaire format using an institutional account to ensure the security and protection of data. Each educational institution was responsible for managing and promoting the completion of the online questionnaires, which were distributed by the respective teachers. Consent and assent forms were signed, and the online questionnaires were completed between May and July of 2021, enabling the execution of the study.

### Variables and Assessment Instruments

*Sleep hygiene indicators:* The Sleep and Rest section (Section 6) of the Questionnaire on Lifestyle and Adolescence: *Sleep Hygiene* [30]. This instrument provides descriptive characteristics of sleep hygiene and contains six questions; two questions about sleep schedules (what time do you wake up? and what time do you go to bed?) and four questions about sleep quality in adolescents, with response options of yes, no, and sometimes (In the mornings, do you wake up feeling tired and would you continue sleeping? Do you have nightmares at night? Do you sleep through the night without waking up? Do you have trouble falling asleep?). Two questions about emotional perception that were unrelated to the study objective were omitted.

The responses obtained from the health hygiene indicators were used as grouping variables to analyze adherence to the Mediterranean diet.

*Adherence to the Mediterranean diet:* The KidMed questionnaire was used to assess adherence to the Mediterranean diet in children and adolescents [31]. The questionnaire consists of 16 items with a scoring system of +1 for positive aspects and/or −1 for negative aspects. The total score is categorized into three levels: ≤3 points indicate low adherence to the Mediterranean diet; 4–7 points indicate a need to improve adherence to the Mediterranean diet; and ≥8 points indicate optimal adherence to the Mediterranean diet. The reliability and reproducibility values are moderate (Cronbach’s alpha = 0.79; 95% CI: 0.71–0.77, and κ = 0.66; 95% CI: 0.45–0.77) [32].

Socio-educational data: Additional data collected included the students’ sex, age, place of residence, educational institution, vulnerability index, and location of the educational institution.

*Data Analysis*: Quantitative variables were presented as mean ± standard deviation, and qualitative variables were presented as absolute frequency and percentage. To determine data distribution and variance homogeneity, the Kolmogorov–Smirnov and Levene tests were used, respectively. The analyzed variables exhibited normal distribution and variance homogeneity, thus warranting the application of parametric statistical methods. A one-way analysis of variance (ANOVA) was conducted to analyze adherence to the Mediterranean diet according to variables related to sleep hygiene. Post hoc analysis using the Bonferroni test was performed to determine differences between pairs of groups when significant differences were found. Cohen’s d effect size (ES) was calculated through the partial eta squared (η_p_^2^) and qualitatively categorized as small (≤0.02), medium (0.02–0.09), or large (>0.09). Subsequently, a multivariate analysis was conducted using multiple linear regression. The results are presented as β coefficients with their respective 95% confidence intervals, according to sleep hygiene. Waking up after 8:30 a.m., going to bed after midnight, waking up feeling tired, having nightmares, sleeping through the night, and having difficulty falling asleep were considered reference values (ref.). The statistical analyses were incrementally adjusted. Model 0: unadjusted model, Model 1: model adjusted for socio-educational variables of the students and educational institutions (sex, age, place of residence, vulnerability index of the educational institution, geographic location of the educational institution). All analyses were performed using SPSS version 26. A significance level of *p* < 0.05 was considered for all analyses.

## 3. Results

Table 1 presents the sociodemographic, health, and educational characteristics of the students. It was observed that the students had an average age of 13.5 years and were mostly females (56.6%) and lived in rural areas (63%). Most students were found to require improvement in adherence to the Mediterranean diet or had a low-quality Mediterranean diet (69.4%). Additionally, it was observed that all analyzed educational centers had a vulnerability index of at least 90%.

In Table 2, the characteristics related to the sleep hygiene of the analyzed students are presented. It was observed that many students woke up after 8:30 a.m. (40%) and went to bed after 11:00 p.m. (46%). On the other hand, a high percentage of students indicated that they sometimes woke up tired and would like to continue sleeping (63.8%), did not sleep through the night without waking up (46%), and had difficulty falling asleep (42.6%), suggesting that a high percentage of students perceive having habits related to poor sleep hygiene.

Table 3 and Figure 1 depict the adherence to the Mediterranean diet of the students according to sleep hygiene indicators. A one-way ANOVA analysis revealed significant differences in the KidMed questionnaire scores based on the time students woke up (F(4;264) = 5.154; *p* = 0.001; η_p_^2^ = 0.073), the time they went to bed (F(4;264) = 8.428; *p* < 0.0001; η_p_^2^ = 0.115), morning tiredness (F(2;264) = 13.338; *p* < 0.0001; η_p_^2^ = 0.092), the presence of nighttime nightmares (F(2;264) = 3.487; *p* = 0.032; η_p_^2^ = 0.026), and difficulty falling asleep (F(2;264) = 7.437; *p* = 0.001; η_p_^2^ = 0.054). Subsequently, a more detailed analysis of group comparisons showed that students who woke up after 8:30 a.m. (*p* < 0.05) and those who went to bed after midnight (*p* < 0.05) had lower adherence to the Mediterranean diet compared to students who woke up before 8:30 a.m. and went to bed before midnight, respectively. Similarly, it was found that students who woke up tired (*p* < 0.05) had lower adherence to the Mediterranean diet than students who did not wake up tired or who did so occasionally. Additionally, it was observed that students who did not have difficulty falling asleep (*p* < 0.05) had better adherence to the Mediterranean diet than students who experienced sleep difficulties or occasionally had such problems. These results suggest that students with indicators of poor sleep hygiene indicators have lower adherence to the Mediterranean diet compared to their peers with good sleep hygiene indicators.

Subsequently, the association between variables related to sleep hygiene and adherence to the Mediterranean diet was analyzed, and whether this association was independent of adjustment variables (Model 1 adjusted). A linear regression analysis was conducted, first with an unadjusted model (Model 0) and then with an adjusted model (Model 1) for socio-educational variables of the students and educational institutions (gender, age, students’ place of residence, vulnerability index of the educational institution, geographic location of the educational institution). In the unadjusted model, it was observed that students who habitually woke up before 8:30 a.m. had significantly higher adherence to the Mediterranean diet than students who tended to wake up after 8:30 a.m. (Before 7:00 h βᵢ = 1.18; *p* = 0.034; Between 7:00 and 7:30 h βᵢ = 1.57; *p* = 0.001; Between 7:30 and 8:00 h βᵢ = 1.43; *p* = 0.003; Between 8:00 and 8:30 h βᵢ = 1.57; *p* = 0.001). Regarding bedtime, it was found that students who habitually went to bed before midnight also had higher adherence to the Mediterranean diet compared to those who usually went to bed after midnight (Before 21:00 h βᵢ = 2.26; *p* = 0.008; Between 21:00 and 22:00 h βᵢ = 2.62; *p* = 0.000; Between 22:00 and 23:00 h βᵢ = 1.93; *p* = 0.000; Between 23:00 and 24:00 h βᵢ = 1.15; *p* = 0.015). On the other hand, it was evident that students who indicated not waking up tired or doing so occasionally had significantly higher levels of adherence to the Mediterranean diet than students who regularly woke up tired (not waking up tired βᵢ = 2.41; *p* = 0.000; Sometimes waking up tired βᵢ = 1.72; *p* = 0.000). Additionally, students who reported no difficulties falling asleep had higher adherence to the Mediterranean diet compared to students who had difficulty falling asleep (no difficulties falling asleep βᵢ = 1.57; *p* = 0.000). Similar results were found in a model adjusted for socio-educational variables of the students and educational institutions. These results suggest that students with better sleep hygiene indicators have better adherence to a healthy diet, and these results are independent of adjustment variables such as gender, age, students’ place of residence, vulnerability index of the educational institution, and geographic location of the educational institution (Table 4).

## 4. Discussion

The main findings of this study suggest that most schoolchildren need to improve their adherence to the Mediterranean diet or have a low-quality Mediterranean diet. Additionally, they presented problems related to sleep hygiene, such as late bedtime and waking up, feeling tired and wanting to keep sleeping, waking up during the night, and difficulty falling asleep. Furthermore, it was evident that schoolchildren reporting better sleep hygiene showed better adherence to the Mediterranean diet compared to those with poorer sleep hygiene.

### 4.1. How Do These Findings Relate to the Existing Literature?

The Mediterranean diet is a widely recognized healthy dietary pattern due to its association with the prevention of noncommunicable chronic diseases [33,34]. Central Chile, the place where this study was developed, has a Mediterranean-like setting with plant and animal food production and availability patterns comparable to those present in countries located around the Mediterranean Sea [35]. Our findings demonstrated that 30.6% of schoolchildren have good adherence to the Mediterranean diet, which is consistent with similar research assessing this indicator in Chilean children and adolescents [36]. Despite the geographical similarities between Chile and Mediterranean countries, which would facilitate the implementation of this diet, adherence remains low [37]. The dietary habits of children and adolescents in Chile are characterized by high consumption of refined sugars and fats, higher ingestion of risky foods, limited access to protective foods, and a higher prevalence of overweight and obesity among less privileged groups, either in rural contexts or lower socioeconomic levels, according to the latest national food consumption survey [38].

Furthermore, a high percentage of schoolchildren and adolescents in our study exhibited sleep hygiene problems, which has been demonstrated in previous studies documenting changes in sleep patterns throughout their development [39]. The reduction in sleep duration, especially daytime somnolence, not only poses difficulties in carrying out daily activities but also leads to behavioral disturbances and an increased risk of cardiovascular disease [40,41], changes in body composition (fat mass) [42], as well as impaired learning capacity [43]. The rise in inadequate sleep among schoolchildren has been associated with the use of screens and electronic devices, which have been shown to directly influence sleep duration and quality in schoolchildren and adolescents [44].

The association between sleep duration and weight has been widely supported [45,46]. However, a meta-analysis reported differences in the study designs, particularly in how sleep duration was considered, as some studies regarded it as a cause and others because of obesity [38]. In our study, we observed sleep hygiene indicators in schoolchildren (bedtime, sleep duration, difficulties in falling asleep, nightmares, morning tiredness) and how a healthy dietary pattern such as the Mediterranean diet could impact their sleep. It was evident that schoolchildren with better sleep hygiene exhibited higher adherence to the Mediterranean diet. This finding is expected since sleep deprivation has been associated with increased calorie intake and unhealthy eating habits, as well as increased snacking and the number of meals consumed per day [47]. Moreover, other studies have explained this association by the high content of antioxidants and the anti-inflammatory properties of the diet [17,48], which decrease the secretion of cytokines associated with sleep deprivation [49].

### 4.2. Limitations of the Study

Although the findings regarding the relationship between sleep variables and diet quality provide an initial understanding of how we can improve components of the lifestyles of schoolchildren and adolescents, the study has several limitations. Due to its cross-sectional design, the study only allowed us to understand how the variables were present at a specific moment and associate them, limiting the control and monitoring of behavior throughout the school year. Additionally, the study design does not allow us to determine the causes of poor sleep hygiene or poor adherence to the Mediterranean diet. Furthermore, self-report questionnaires were used in an online format to assess the variables, which does not guarantee the accuracy of the adolescents’ responses, as they may have different motivations when answering the questionnaires, despite ensuring confidentiality and anonymity. For future investigations concerning sleep quantity and quality analysis, the incorporation of more objective sleep measures, such as actigraphy, is recommended. Furthermore, the inclusion of data pertaining to medical history, prevailing health status, and nutritional indicators (e.g., weight, height, or body mass index) was not undertaken. This additional information could have potentially facilitated a more refined adjustment of the proposed model.

### 4.3. What Are the Contributions and Practical Implications of This Study?

The findings of this research support current evidence and contribute to the existing body of research in Latin America suggesting an association between sleep problems and dietary patterns in schoolchildren. To date, this study represents the most up-to-date evidence that shows that a high percentage of students from rural educational centers in southern Chile (52.8%) need to improve their adherence to the Mediterranean diet. In addition, this study shows that a high percentage of students need to improve sleep hygiene indicators (such as late bedtime and waking up, feeling tired and wanting to keep sleeping, waking up during the night, and difficulty falling asleep). Furthermore, this study found an association between poor sleep hygiene such as difficulty waking up and falling asleep, and lower adherence to the Mediterranean diet among schoolchildren attending rural public schools in a province in southern Chile. This research highlights the importance of systematically analyzing and monitoring schoolchildren’s sleep and dietary habits to early detect problems and potentially prevent their implications on health. Additionally, these findings can be considered by professionals and administrators in educational and healthcare settings to reflect on the potential negative impact of poor sleep hygiene and diet on the health of schoolchildren.

### 4.4. Future Lines of Research

It is encouraged to investigate the interaction among the studied variables, as well as the association between current lifestyle habits. For instance, the physical space where individuals sleep and eat, the use of electronic devices, low levels of physical activity, and eating habits or routines. Recent studies conducted in Chile have already begun to yield helpful insights on the association between unhealthy habits in childhood [7,29,50].

## 5. Conclusions

Our study concludes that having poor sleep hygiene is associated with lower adherence to the Mediterranean diet in adolescent schoolchildren. Additionally, most students exhibit poor diet quality or need to improve their adherence to the diet, and they report having sleep hygiene problems, regardless of sociodemographic variables. The results highlight the importance of monitoring these variables in schoolchildren and promoting healthy lifestyle habits within the educational community. Further studies investigating the evolution of these variables and the development of intervention programs are needed.

## Figures and Tables

**Figure 1 children-10-01499-f001:**
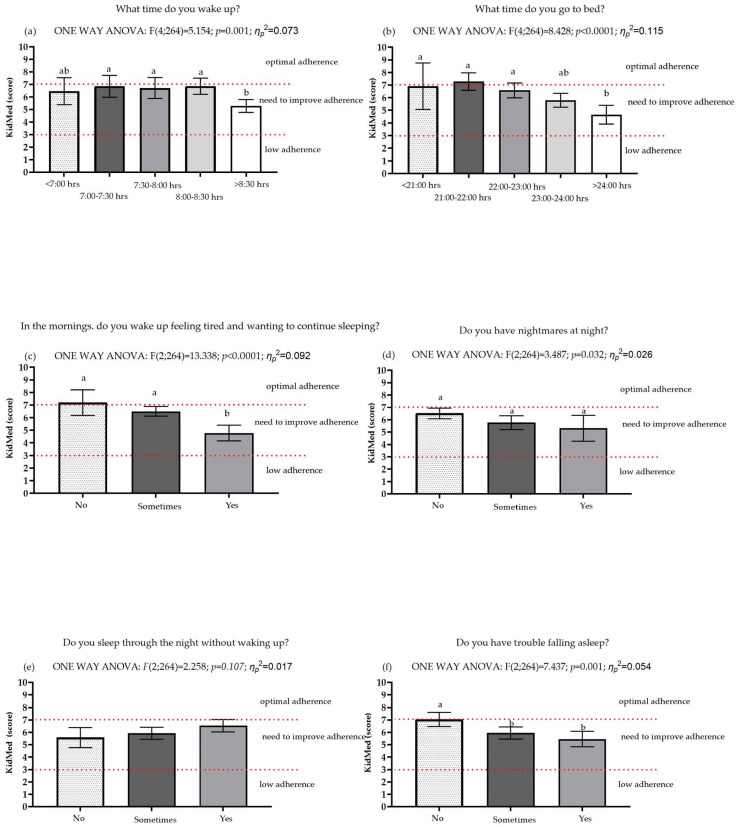
Adherence to the Mediterranean diet among the grouping variables. Note: The statistical analysis was conducted using one-way ANOVA. The total score is categorized into three levels: ≤3 points indicate low adherence to the Mediterranean diet; 4–7 points indicate a need to improve adherence to the Mediterranean diet; and ≥8 points indicate optimal adherence to the Mediterranean diet. Different letters ab in the same row indicate significant differences between groups (Post hoc multiple comparison with Bonferroni test). A significance level of *p* < 0.05 was considered for all analyses.

**Table 1 children-10-01499-t001:** Characteristics of the students.

Sociodemographic Variables of the Students	Mean	SD
Age (years)	13.5	1.8
Gender	Frequency	Percentage
*Male*	115	43.4%
*Female*	150	56.6%
Residence location of the students		
*Out of town*	167	63%
*In town*	98	37%
KidMed Classification		
*Optimal adherence to the Mediterranean diet*	81	30.6%
*Needs improvement in adherence to the Mediterranean diet*	140	52.8%
*Low-quality Mediterranean diet*	44	16.6%
Characterization variables of the educational establishments	Frequency	Percentage
*Educational establishment n°1*	3	1.1%
*Educational establishment n°2*	13	4.9%
*Educational establishment n°3*	19	7.2%
*Educational establishment n°4*	4	1.5%
*Educational establishment n°5*	15	5.7%
*Educational establishment n°6*	10	3.8%
*Educational establishment n°7*	11	4.2%
*Educational establishment n°8*	13	4.9%
*Educational establishment n°9*	35	13.2%
*Educational establishment n°10*	142	53.6%
Vulnerability Index (VI) of the educational establishment		
*90–95% VI*	78	29.4%
*96–100% VI*	187	70.6%
Educational establishment by geographic location		
*Out of town*	69	26%
*In town*	196	74%

Note: The quantitative variables were presented as mean ± standard deviation, while the qualitative variables were presented as absolute frequency and percentage. VI stands for Vulnerability Index. n = 265.

**Table 2 children-10-01499-t002:** Sleep hygiene-related characteristics of the students.

Variables Related to Sleep Hygiene	Frequency	Percentage
What time do you wake up?		
*Before 7:00 h*	28	10.6%
*Between 7:00 and 7:30 h*	40	15.1%
*Between 7:30 and 8:00 h*	42	15.8%
*Between 8:00 and 8:30 h*	49	18.5%
*After 8:30 h*	106	40.0%
What time do you go to bed?		
*Before 21:00 h*	11	4.2%
*Between 21:00 and 22:00 h*	58	21.9%
*Between 22:00 and 23:00 h*	74	27.9%
*Between 23:00 and 24:00 h*	70	26.4%
*After 24:00 h*	52	19.6%
In the mornings. do you wake up feeling tired and wanting to continue sleeping?		
*No*	31	11.7%
*Sometimes*	169	63.8%
*Yes*	65	24.5%
Do you have nightmares at night?		
*No*	151	57.0%
*Sometimes*	92	34.7%
*Yes*	22	8.3%
Do you sleep through the night without waking up?		
*No*	117	44.2%
*Sometimes*	122	46.0%
*Yes*	26	9.8%
Do you have trouble falling asleep?		
*No*	83	31.3%
*Sometimes*	113	42.6%
*Yes*	69	26.0%

Note: Qualitative variables were presented as absolute frequency and percentage. n = 265.

**Table 3 children-10-01499-t003:** Adherence to the Mediterranean diet according to variables related to sleep hygiene.

Adherence to the Mediterranean Diet (KIDMED)											
Mean [95% CI]	Mean [95% CI]	Mean [95% CI]	Mean [95% CI]			Mean [95% CI]			One-Way ANOVA		
What time do you wake up?									F	*p*-Value	η_p_^2^
Before 7:00 h	Between 7:00 and 7:30 h	Between 7:30 and 8:00 h	Between 8:00 and 8:30 h			After 8:30 h					
6.46 [5.39;7.54] ^ab^	6.85 [5.98;7.72] ^a^	6.71 [5.88;7.55] ^a^	6.86 [6.21;7.5] ^a^			5.28 [4.77;5.79] ^b^			5.154	0.001	0.073
What time do you go to bed?											
Before 21:00 h	Between 21:00 y 22:00 h	Between 22:00 and 23:00 h	Between 23:00 and 24:00 h			After 24:00 h					
6.91 [5.07;8.75] ^a^	7.28 [6.58;7.97] ^a^	6.58 [5.99;7.17] ^a^	5.8 [5.25;6.35] ^ab^			4.65 [3.91;5.4] ^b^			8.428	<0.0001	0.115
In the mornings. do you wake up feeling tired and wanting to continue sleeping?											
No	Sometimes	Yes							F	*p*-Value	η_p_^2^
7.19 [6.18;8.21] ^a^	6.5 [6.12;6.89] ^a^	4.78 [4.16;5.41] ^b^							13.338	<0.0001	0.092
Do you have nightmares at night?											
No	Sometimes	Yes									
6.52 [6.09;6.95] ^a^	5.77 [5.21;6.33] ^a^	5.32 [4.27;6.36] ^a^							3.487	0.032	0.026
Do you sleep through the night without waking up?											
No	Sometimes	Yes									
5.58 [4.77;6.39]	5.93 [5.44;6.42]	6.54 [6.04;7.04]							2.258	0.107	0.017
Do you have trouble falling asleep?											
No	Sometimes	Yes									
7.04 [6.47;7.61] ^a^	5.95 [5.46;6.44] ^b^	5.46 [4.84;6.09] ^b^							7.437	0.001	0.054

Note: The statistical analysis was conducted using one-way ANOVA. Different letters ab in the same row indicate significant differences between groups (Post hoc multiple comparison with Bonferroni test). A significance level of *p* < 0.05 was considered for all analyses.

**Table 4 children-10-01499-t004:** Association between variables related to sleep hygiene and adherence to the Mediterranean diet.

		Adherence to the Mediterranean Diet (KIDMED)			
	**βᵢ [CI 95%]**	**βᵢ [CI 95%]**	**βᵢ [CI 95%]**	**βᵢ [CI 95%]**	**Reference group**
	What time do you wake up?				
Variables	Before 7:00 h	Between 7:00 and 7:30 h	Between 7:30 and 8:00 h	Between 8:00 and 8:30 h	After 8:30 h
*Model 0 (Unadjusted)*	1.18 [0.09;2.27] *	1.57 [0.62;2.52] **	1.43 [0.50;2.37] **	1.57 [0.69;2.46] **	Ref.
*Model 1 (Adjusted)*	1.08 [−0.07;2.23]	1.41 [0.33;2.48] **	1.51 [0.54;2.48] **	1.39 [0.48;2.31] **	Ref.
	What time do you go to bed?				
	Before 21:00 h	Between 21:00 and 22:00 h	Between 22:00 and 23:00 h	1.	After 24:00 h
*Model 0 (Unadjusted)*	2.26 [0.59;3.92] **	2.62 [1.66;3.58] ***	1.93 [1.02;2.83] ***	1.15 [0.23;2.06] *	Ref.
*Model 1 (Adjusted)*	1.91 [0.14;3.68] *	2.41 [1.38;3.44] ***	1.73 [0.78;2.67] ***	1.12 [0.19;2.04] *	Ref.
	**βᵢ [IC 95%]**	**βᵢ [IC 95%]**	**Reference group**		
In the mornings. do you wake up feeling tired and wanting to continue sleeping?					
Variables	No	Sometimes	Yes		
*Model 0 (Unadjusted)*	2.41 [1.31;3.51] ***	1.72 [0.98;2.46] ***	Ref.		
*Model 1 (Adjusted)*	2.23 [1.10;3.35] ***	1.67 [0.93;2.42] ***	Ref.		
Do you have nightmares at night?					
	No	Sometimes	Yes		
*Model 0 (Unadjusted)*	1.20 [0.01;2.40] *	0.45 [−0.79;1.70]	Ref.		
*Model 1 (Adjusted)*	1.16 [−0.04;2.36]	0.47 [−0.78;1.71]	Ref.		
Do you sleep through the night without waking up?					
	No	Sometimes	Yes		
*Model 0 (Unadjusted)*	0.96 [−0.18;2.10]	0.35 [−0.79;1.49]	Ref.		
*Model 1 (Adjusted)*	0.92 [−0.23;2.07]	0.41 [−0.73;1.55]	Ref.		
Do you have trouble falling asleep?					
	No	Sometimes	Yes		
*Model 0 (Unadjusted)*	1.57 [0.73;2.41] ***	0.48 [−0.31;1.27]	Ref.		
*Model 1 (Adjusted)*	1.50 [0.65;2.35] **	0.57 [−0.24;1.37]	Ref.		

Note: The data are presented as β coefficients and their respective 95% confidence intervals (CI) for sleep hygiene variables. The statistical analyses were conducted using multiple linear regression analysis. Waking up after 8:30 a.m., sleeping after midnight, waking up tired, having nightmares, sleeping through the night, and having difficulty falling asleep were considered to be reference values (ref.). The statistical analyses were incrementally adjusted. Model 0: unadjusted model, Model 1: adjusted model for socio-educational variables of the students and educational establishments (gender, age, place of residence of the students, vulnerability index of the educational establishment, geographic location of the educational establishment). *** = differences are significant with a *p*-value < 0.001, ** = differences are significant with a *p*-value < 0.01, * = differences are significant with a *p*-value < 0.05.

## Data Availability

Data will be made available upon request.

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
