# Peer review of "Low Sleep Hygiene Is Associated with Less Adherence to the Mediterranean Diet in Chilean Schoolchildren from Rural Public Schools—A Cross-Sectional Study"

_children, 2023, doi:10.3390/children10091499_

Round 1

Reviewer 1 Report

Low sleep hygiene is associated with less adherence to the
Mediterranean diet in Chilean schoolchildren from rural public schools. A cross-sectional study (children-2496863)
(Review)

Main message of the article

The article by Zapata-Lamana and colleagues explores the relationship between sleep habits and adherence to the Mediterranean diet in a sample of 265 students from Chile. Results show that a large part of the sample has a low level of adherence to the Mediterranean diet. Furthermore, a high percentage of participants have behaviors related to poor sleep hygiene. Furthermore, it was observed that students that tend to get up after 8:30 a.m., fall asleep after 12:00 a.m., wake up tired, and have trouble falling asleep have lower levels of adherence to the Mediterranean diet.

General Judgment Comments

The article is clearly written. The title is clear, but the abstract needs more information in regard to the theoretical background, rationale of the study, aims, and results. The authors need to clarify the way in which the needed sample was determined. Furthermore, assumptions of ANOVA should be checked before running the test. If the data violate the assumptions, the researchers should adopt non-parametric tests. In the study, it is not clear which test was used for post-hoc analyses. Results should be reported in the standard format and effect sizes should be added to the manuscript.

For these reasons, I recommend the manuscript to undergo Major Revision.

Major Issues

  • -  Abstract: the theoretical background and the rationale for conducting this study are missing.

  • -  Abstract: results are not reported for the comparisons among groups defined by the sleep patterns.

  • -  Abstract: the concluding sentence seem quite generic.

  • -  Methods: How was the sample size determined?

  • -  Methods: how and why were questionnaires selected? In which language were the questionnaires written? What is the Cronbach’s alpha within scales?

  • -  Methods: were the assumptions of the ANOVA tested before conducting it? What are the groups between which

    adherence to the Mediterranean diet was assessed?

  • -  Methods: “Post hoc analysis using the Bonferroni test was performed to determine differences between pairs of groups when significant differences were found.” Bonferroni is not a test but a correction for the alpha level. What post’hoc test did the author conduct?

  • -  Results: results are not reported in the standard format. Furthermore, please add the effect sizes.

  • -  Lines 190-200: Bonferroni is not a test, but a correction of the alpha level when conducting multiple comparisons. In

    this paragraph, no result is reported. Please add the results following the standard format.

  • -  Please add a figure in which you show the adherence to the Mediterranean diet across different groups. I would suggest

    dividing the figure into panels, with each panel representing the specific sleep variable that was considered.

  • -  Lines 208-228: no result is presented in the main text. Please add the results in the standard format.

    Minor Issues

  • -  Abstract: “Non-experimental, analytical, cross-sectional study”. This sentence alone does not mean much. Please rephrase it and use verbs.

  • -  Introduction: “Sleep hygiene is intended to control the conditions that influence the sleep of children and adolescents, including interventions to promote sleep hygiene (2)”. This sentence is unclear. Please rephrase it.

  • -  Introduction: “Thus, it may be relevant to evaluate how a healthy dietary pattern, such as the Mediterranean diet, which has evidence of its protective power against inflammation [...]”. Here the authors need to provide evidence by referring to existing works. The same applies to the sentence “In recent years, sleep hygiene has been considered a variable of special interest associated with nutrition.”

  • -  Introduction: “In Latin American countries like ours” should become “in Latin American countries like Chile”.

  • -  Limitations of the study: when discussing the limits of using self-report questionnaires, the authors could suggest the

    use of more objective sleep measures (e.g., actigraphy) in future studies.

    Final comments

    For these reasons, I recommend the manuscript to undergo Major Revision. 

Author Response

Main message of the article

The article by Zapata-Lamana and colleagues explores the relationship between sleep habits and adherence to the Mediterranean diet in a sample of 265 students from Chile. Results show that a large part of the sample has a low level of adherence to the Mediterranean diet. Furthermore, a high percentage of participants have behaviors related to poor sleep hygiene. Furthermore, it was observed that students that tend to get up after 8:30 a.m., fall asleep after 12:00 a.m., wake up tired, and have trouble falling asleep have lower levels of adherence to the Mediterranean diet.

Answer: We appreciate the well-constructed abstract provided for the manuscript.

General Judgment Comments

The article is clearly written. The title is clear, but the abstract needs more information in regard to the theoretical background, rationale of the study, aims, and results. The authors need to clarify the way in which the needed sample was determined. Furthermore, assumptions of ANOVA should be checked before running the test. If the data violate the assumptions, the researchers should adopt non-parametric tests. In the study, it is not clear which test was used for post-hoc analyses. Results should be reported in the standard format and effect sizes should be added to the manuscript.

Answer: Thank you so much for the comments. We are confident that your insights will contribute to the enhancement of this manuscript. Below, you will find our responses addressing each of the suggestions provided.

MAYOR ISSUES

  • Abstract:
    • the theoretical background and the rationale for conducting this study are missing.

Answer: The recommended content has been incorporated as advised

  • results are not reported for the comparisons among groups defined by the sleep patterns.

Answer: The abstract delineates descriptive results followed by intergroup comparative results. We will undertake a revision of these sections to enhance their clarity and facilitate comprehension.

  • the concluding sentence seem quite generic.

Answer: The conclusion has been rephrased to make it more specific.

General answer: The reviewer's feedback is acknowledged, and their comments have been taken into careful consideration. However, it is important to note that the journal's guidelines stipulate a maximum 200 word for the abstract. Currently, the abstract comprises 248 words. While we have incorporated some of the reviewer's suggestions, the word limit presents a real challenge in incorporating all the proposed changes. For further reference, please find attached the provided instructions for authors, and the link to the instructions for authors is also provided https://www.mdpi.com/journal/children/instructions

  • Methods:
    • How was the sample size determined?

Answer: There was no sample size calculation, because many schoolchildren did not complete the questionnaires, we could not have a representative sample, which hindered the establishment of a representative sample. As a result, a non-probabilistic sample of voluntary subjects was utilized.

  • How and why were questionnaires selected? In which language were the questionnaires written? What is the Cronbach’s alpha within scales?

Answer: An analysis of suitable instruments was conducted to identify appropriate instruments for implementation within a virtual setting. This evaluation was conducted within a geographically intricate community characterized by a substantial rural population, coupled with constrained and intermittent internet connectivity. The suitability of instruments was determined based on criteria encompassing their comprehensiveness, linguistic accessibility, and ease of comprehension. It is noteworthy that both questionnaires employed in this study were composed in Spanish and administered accordingly.

The instrument to evaluate sleep hygiene is a section of an instrument that evaluates life habits and adolescence. It is noteworthy that this specific section lacks independent validation, a limitation that we acknowledge within the context of our discussion. Conversely, in the context of the instrument utilized to appraise adherence to the Mediterranean diet (KIDMED), the pertinent information that had been requested has been duly incorporated.

  • were the assumptions of the ANOVA tested before conducting it? What are the groups between which adherence to the Mediterranean diet was assessed?

Answer: The assumptions of normality and homogeneity of variance were evaluated, the variables analyzed had a normal distribution and homogeneity of variances, for this reason the ANOVA test was used. This information is now clarified in the statistical analysis section.

Adherence to the Mediterranean diet was analyzed based on the responses obtained in the health hygiene indicators that are presented in the methodology. Now, in the new version of the manuscript it is explained to clarify this point.

  • Post hoc analysis using the Bonferroni test was performed to determine differences between pairs of groups when significant differences were found.” Bonferroni is not a test but a correction for the alpha level. What post’hoc test did the author conduct?

Answer: We acknowledge the comment provided by the reviewer; however, it appears that there might be a misinterpretation regarding the application of the Bonferroni test. Notably, the Bonferroni test is commonly employed as a post hoc multiple comparison test within the SPSS program, particularly when the assumption of equal variances is presumed

  • Results:
    • results are not reported in the standard format. Furthermore, please add the effect sizes.
    • Lines 190-200: Bonferroni is not a test, but a correction of the alpha level when conducting multiple comparisons. In this paragraph, no result is reported. Please add the results following the standard format.
    • Please add a figure in which you show the adherence to the Mediterranean diet across different groups. I would suggest dividing the figure into panels, with each panel representing the specific sleep variable that was considered.
    • Lines 208-228: no result is presented in the main text. Please add the results in the standard format.

Answer: We have incorporated the effect size into the analysis, and a figure has been devised to illustrate the variance in adherence to the Mediterranean diet among distinct groups. Furthermore, the results section has been restructured to adhere to a standardized format, enhancing clarity.

MINOR ISSUES

  • Abstract:
    • “Non-experimental, analytical, cross-sectional study”. This sentence alone does not mean much. Please rephrase it and use verbs.

Answer: The sentence was modified.

  • Introduction:
    • “Sleep hygiene is intended to control the conditions that influence the sleep of children and adolescents, including interventions to promote sleep hygiene (2)”. This sentence is unclear. Please rephrase it.

Answer: Thanks for the comment. The sentence in question has been rectified, and a new reference has been seamlessly integrated as advised.

“Thus, it may be relevant to evaluate how a healthy dietary pattern, such as the Mediterranean diet, which has evidence of its protective power against inflammation [...]”. Here the authors need to provide evidence by referring to existing works. The same applies to the sentence “In recent years, sleep hygiene has been considered a variable of special interest associated with nutrition.”

Answer: Thank you very much, the sentence has been rephrased, and additional references have been incorporated to substantiate its content.

 “In Latin American countries like ours” should become “in Latin American countries like Chile”.

Answer: Thank you very much, the amendment was made.

  • Limitations of the study:
    • When discussing the limits of using self-report questionnaires, the authors could suggest the use of more objective sleep measures (e.g., actigraphy) in future studies.

Answer: Thank you very much, suggested comment added.

Final comments

For these reasons, I recommend the manuscript to undergo Major Revision. 

Reviewer 2 Report

I have a number of further comments to improve the manuscript:

·      What constitutes the Mediterranean diet? Why is it crucial to adhere to this dietary pattern, and how does it relate to sleep? It is essential to include a comprehensive explanation to address these questions thoroughly in the Introduction session.

·      When choosing participants, it is crucial to consider factors such as weight and medical history. However, in this research, there seems to be no mention of how these factors were addressed or taken into account, as I couldn't locate any relevant information in the manuscript.

·      This study reveals the most up-to-date evidence of a relationship between poor sleep hygiene and lower adherence to the Mediterranean diet among schoolchildren attending rural public schools in a province in southern Chile. How can we ascertain the veracity of this matter? It is essential to provide an explanation grounded findings based on the table's result and a statistical method employed.

·      The primary objective of this study centers on Chilean schoolchildren attending rural public schools. However, in Table 1, it is observed that the educational establishments listed are predominantly urban.  The impact of this rural bias on the research findings needs to be elucidated.

Minor editing of English language required.

Author Response

I have a number of further comments to improve the manuscript:

  • What constitutes the Mediterranean diet? Why is it crucial to adhere to this dietary pattern, and how does it relate to sleep? It is essential to include a comprehensive explanation to address these questions thoroughly in the Introduction session.

Answer: Thank you very much for the comment. A comprehensive sentence has been included to provide a description of the Mediterranean diet and elucidate its connection with sleep patterns.

  • When choosing participants, it is crucial to consider factors such as weight and medical history. However, in this research, there seems to be no mention of how these factors were addressed or taken into account, as I couldn't locate any relevant information in the manuscript.

Answer: We agree with the reviewer that medical information and nutritional status are relevant to this study. Unfortunately, this information was not requested and is added as a limitation of the study.

  • This study reveals the most up-to-date evidence of a relationship between poor sleep hygiene and lower adherence to the Mediterranean diet among schoolchildren attending rural public schools in a province in southern Chile. How can we ascertain the veracity of this matter? It is essential to provide an explanation grounded findings based on the table's result and a statistical method employed.

Answer: We agree with the feedback provided. The sentence has been rephrased to align with the results as depicted in the tables presented.

  • The primary objective of this study centers on Chilean schoolchildren attending rural public schools. However, in Table 1, it is observed that the educational establishments listed are predominantly urban. The impact of this rural bias on the research findings needs to be elucidated.

Answer: We appreciate your clarification. I apologize for any confusion. The concept has been rectified accordingly to accurately reflect the geographical context within the tables, distinguishing between children and educational establishments situated within the urban center and those situated in the outskirts of "El Carmen," the rural community in Chile. (https://es.wikipedia.org/wiki/El_Carmen_(comuna)

Round 2

Reviewer 1 Report

Authors have revised the article as requested (more or less)